# Risk Factors for Hypertension in Hospitalised Patient Mortality with Laboratory-Confirmed SARS-CoV-2: A Population-Based Study in Limpopo Province, South Africa

Peter M. Mphekgwana [1,*], Sogo F. Matlala [2], Takalani G. Tshitangano [3], Naledzani J. Ramalivhana [4] and Musa E. Sono-Setati [4,5]

1   Department of Research Administration and Development, University of Limpopo, Polokwane 0727, South Africa
2   Department of Public Health, University of Limpopo, Polokwane 0727, South Africa
3   Department of Public Health, University of Venda, Polokwane 0950, South Africa
4   Limpopo Department of Health, College Ave, Hospital Park, Polokwane 0699, South Africa
5   School of Medicine, Department of Public Health Medicine, University of Limpopo, Polokwane 0727, South Africa
*   Correspondence: peter.mphekgwana@ul.ac.za; Tel.: +27-15-268-3982

**Abstract:** The coronavirus disease (COVID-19) pandemic has recently impacted and destabilised the global community. The healthcare systems of many countries have been reported to be partially or entirely interrupted. More than half of the countries surveyed (53%) have partially or completely disrupted hypertension treatment services. A population-based retrospective cohort study approach was used to determine the prevalence of hypertension and related risk factors for mortality in COVID-19 hospitalised patients in the Limpopo Province, South Africa. Hierarchical logistic regression was applied to determine the determinants of hypertension. Sixty-nine percent (69%) of mortality among individuals with laboratory-confirmed SARS-CoV-2 were elderly persons aged 60 years and above admitted to a person under investigation (PUI) ward (52%), and 66% had hypertension. Among the hospitalised COVID-19 patients who died, prominent risk factors for hypertension were advanced age, the presence of co-morbidities, such as diabetes and HIV/AIDS. There was no evidence to establish a link between hypertension and COVID-19 case severity. More cohort and systematic studies are needed to determine whether there is a link between hypertension and COVID-19 case severity.

**Keywords:** COVID-19; mortality; SARS-CoV-2; hypertension; hospitalised; patients

## 1. Introduction

The coronavirus disease (COVID-19) pandemic recently impacted and destabilised the global community. As of 28 January 2022, there were 360 million confirmed cases of COVID-19, including 5 million deaths reported worldwide [1]. The increasing burden of COVID-19 cases and deaths resulted in countries implementing national lockdowns, quarantine and other restrictions in an effort to contain the spread of the disease [2]. The pandemic has negatively influenced the economies, social welfare, education and healthcare systems of many nations [3,4]. In South Africa, the first case of SARS-CoV-2 infection was recorded early in March 2020 and the country soon became the most affected country in Africa [5]. As of 28 January 2022, there have been 3 million infections and 94,651 COVID-19 related deaths in South Africa [5].

Recent studies have reported that the most common symptoms of COVID-19 at the onset of illness were fever, cough, fatigue, dyspnea and headache [6]. A study showed that taste and smell disorders, as well as diarrhoea, were later also important symptoms of COVID-19 [7]. However, symptoms of COVID-19 differ from country to country and again from person to person, because symptoms are a result of an individual's immune

response [7,8]. Clinical and risk factors for mortality of adult inpatients with COVID-19 have recently been explored in a large body of studies. These studies revealed that most infected patients had an underlying disease, primarily hypertension, cardiovascular disease and malignancy [8]. Another report further showed that diabetes was also more common in patients who tested positive for COVID-19 [9].

The healthcare systems of many countries have been reported to be partially or entirely interrupted. More than half of the countries surveyed (53%) have partially or completely disrupted hypertension treatment services; 49% have disrupted services for diabetes and diabetes-related complications; 42% have disrupted services for cancer treatment; and 31% have disrupted services for cardiovascular emergencies [10]. Hypertension is responsible for millions of deaths worldwide and has been identified as a significant risk factor for the development of severe COVID-19 [11]. Maintaining basic services is critical in order to minimise the consequences of hypertension and cardiovascular disease. Individuals with hypertension and cardiovascular disease are more likely to develop severe disease and die as a result of COVID-19 [12].

Most recent studies investigated the association between hypertension and COVID-19 [8,11,12]. However, we are not aware of any studies that attempted to establish the prevalence of hypertension and related risk factors for mortality of hospitalised COVID-19 patients in Limpopo Province, South Africa. In the present study, we determined the prevalence of hypertension in the mortality of hospitalised COVID-19 patients and investigated the factors associated with hypertension among the study population.

## 2. Materials and Methods

### 2.1. Study Setting, Period and Design

The study was carried out in Limpopo Province, South Africa. Limpopo Province is the northernmost province of South Africa, consisting of five districts (Capricorn, Mopani, Sekhukhune, Vhembe and Waterberg) divided into 25 local municipalities with an estimated population of 5 million. A population-based retrospective cohort study approach was used to determine the prevalence of hypertension and related risk factors for mortality of COVID-19 hospitalised patients in the Limpopo Province, South Africa. The study used secondary data collected and captured by the Limpopo Department of Health (LDoH). It involved 969 hospitalised patients with clinical syndromes highly suspicious for COVID-19 referred to hospitals in the province from March 2020 to February 2021. Patients were followed from the day of admission at the referred hospital until the occurrence of a laboratory-confirmed COVID-19-related death. The study was approved by the University of Limpopo Research Ethics Committee (TREC/293/2021: IR) and the Limpopo Province Health Research Committee (LP_2021-11-017).

### 2.2. Statistical Analysis

All data analyses were performed using STATA software (Stata 9.0, StataCorp, College Station, TX, USA) and R software. Baseline characteristics of all patients with COVID-19 were expressed as frequencies and percentages. The statistical significance of sex group differences was tested using the chi-square test and a hierarchical logistic regression was applied to determine the determinants of hypertension on hospital patient mortality with laboratory-confirmed SARS-CoV-2.

### 2.3. Data Collection and Validation

This study used mortality audit data to examine all deaths amongst hospitalised COVID-19 patients between the first (16 March 2020 to 31 October 2020) and second (1 November 2020 to 29 February 2021) pandemic waves. As detailed in Tshitangano et al., the LDoH developed the COVID-19 in-patient mortality audit tool which was thoroughly validated by LDoH officials with the help of the WHO COVID-19 provincial support team [13]. The tool was administered and completed by each hospital audit team. The audit was done in a group setting with the help of a senior clinical manager and a nursing

service manager. The data were collected from 41 hospitals in all five districts of Limpopo Province namely, Waterberg, Mopani, Capricorn, Vhembe and Sekhukhune.

## 3. Results

The selected sociodemographic, wards, comorbid conditions and clinical presentations of the patients admitted to referral hospitals in Limpopo Province with clinical syndromes highly suspicious for COVID-19 are presented in Table 1. Sixty-nine percent (69%) were elderly persons aged 60 years and above. Forty-three percent (43%) of these patients were admitted to hospitals in the Capricorn District, followed by 24% in Vhembe District. Overall, there was a significant difference (*p*-value = 0.037) between the sexes of the 56% of patients who were admitted to a person under investigation (PUI) ward. More than 50% of referred patients were on chronic treatment for diabetes and hypertension. There was a higher prevalence of TB among males, at 8%, versus 3% among females. There was a higher prevalence of asthma (7%) and obesity (14%) among female referred patients with clinical syndromes highly suspicious for COVID-19 with a significant difference between the sexes (*p*-value = 0.039 and *p*-value = 0.022, respectively). Most patients presented with fever (55%), cough (81%), shortness of breath (84%) and myalgia/body aches (60%).

**Table 1.** Demographic characteristics of patients with clinical syndromes highly suspicious for COVID-19.

| | Overall | Percentage | Female | Percentage | Male | Percentage | *p*-Value |
|---|---|---|---|---|---|---|---|
| **Age** | | | | | | | 0.986 |
| 20–29 | 10 | 1% | 6 | 1% | 4 | 2% | |
| 30–39 | 62 | 6% | 29 | 5% | 33 | 6% | |
| 40–49 | 101 | 9% | 51 | 9% | 50 | 9% | |
| 50–59 | 172 | 15% | 90 | 15% | 82 | 15% | |
| 60+ | 774 | 69% | 412 | 70% | 362 | 68% | |
| **District** | | | | | | | <0.001 |
| Capricorn | 499 | 43% | 273 | 45% | 226 | 42% | |
| Mopani | 219 | 19% | 116 | 19% | 103 | 19% | |
| Sekhukhune | 94 | 8% | 42 | 7% | 52 | 10% | |
| Vhembe | 279 | 24% | 146 | 24% | 133 | 24% | |
| Waterberg | 57 | 5% | 27 | 4% | 30 | 6% | |
| **Wards** | | | | | | | 0.037 |
| Casualty | 27 | 2% | 10 | 2% | 17 | 3% | |
| General ward | 403 | 37% | 214 | 37% | 189 | 37% | |
| High Care | 23 | 2% | 10 | 2% | 13 | 3% | |
| ICU | 26 | 2% | 17 | 3% | 9 | 2% | |
| PUI ward | 606 | 56% | 325 | 56% | 281 | 55% | |
| **Comorbid conditions** | | | | | | | |
| HIV/AIDS | 141 | 19% | 69 | 18% | 72 | 20% | 0.507 |
| TB | 37 | 5% | 12 | 3% | 25 | 8% | 0.021 |
| COPD | 18 | 3% | 9 | 3% | 9 | 3% | 0.917 |
| Hypertension | 586 | 64% | 319 | 65% | 267 | 62% | 0.344 |
| Diabetes Mellitus | 450 | 52% | 243 | 54% | 207 | 50% | 0.188 |
| Asthma | 35 | 5% | 25 | 7% | 10 | 3% | 0.016 |
| Obesity | 81 | 12% | 52 | 14% | 29 | 9% | 0.022 |
| Cancer | 23 | 4% | 11 | 3% | 12 | 4% | 0.758 |
| Chronic treatment | 493 | 67% | 275 | 70% | 218 | 63% | 0.057 |
| Respiratory distress | 919 | 88% | 483 | 88% | 436 | 89% | 0.188 |
| Mechanical ventilation | 59 | 8% | 33 | 9% | 26 | 8% | 0.692 |

**Table 1.** *Cont.*

| | Overall | Percentage | Female | Percentage | Male | Percentage | *p*-Value |
|---|---|---|---|---|---|---|---|
| **Clinical presentations** | | | | | | | |
| Fever (self-reported) | 266 | 55% | 126 | 53% | 140 | 58% | 0.123 |
| Chills | 148 | 33% | 73 | 34% | 75 | 33% | 0.056 |
| Cough | 650 | 81% | 324 | 81% | 326 | 82% | 0.030 |
| Sore throat | 137 | 32% | 62 | 30% | 75 | 34% | 0.031 |
| Shortness of breath | 714 | 84% | 368 | 83% | 346 | 85% | 0.645 |
| Anosmia | 65 | 16% | 23 | 12% | 42 | 19% | 0.001 |
| Dysgeusia | 83 | 20% | 37 | 19% | 46 | 21% | 0.020 |
| Myalgia/body aches | 374 | 60% | 199 | 62% | 175 | 58% | 0.387 |
| Diarrhoea | 145 | 31% | 74 | 32% | 71 | 30% | 0.071 |
| Chest Pain | 55 | 8% | 29 | 8% | 26 | 8% | 0.973 |
| Loss of appetite | 65 | 9% | 34 | 9% | 31 | 9% | 0.997 |

In Figures, non-COVID-19 patients were viewed as highly suspicious for COVID-19 by medical professionals. These were non-lab-confirmed SARS-CoV-2 patients who were either awaiting test results or had not yet been tested. Of all the patients admitted to referral hospitals in Limpopo Province with clinical syndromes highly suspicious for COVID-19 (969), 887 (92%) were laboratory-confirmed SARS-CoV-2. Sixty-nine percent (69%) of the deaths of individuals with laboratory-confirmed SARS-CoV-2 were elderly persons aged 60 years and above admitted in PUI wards (52%) with hypertension (66%), as shown in Figures 1–4. The mortality rate of individuals with laboratory-confirmed SARS-CoV-2 was higher in the Capricorn District versus nonconfirmed (30%) SARS-CoV-2 (*p*-value < 0.001) patients, followed by the Vhembe District with 20% COVID-19 versus 14% non-COVID-19 (*p*-value < 0.001) patients.

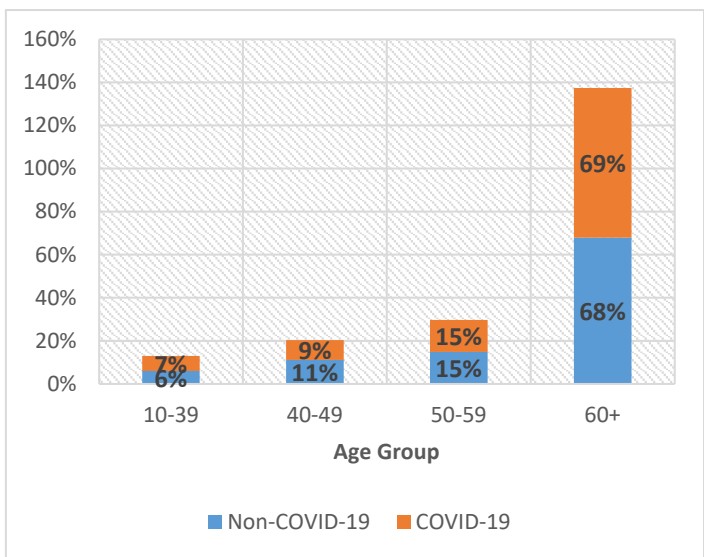

**Figure 1.** Prevalence of mortality among individuals with laboratory-confirmed SARS-CoV-2 according to age group (*p*–value = 0.979).

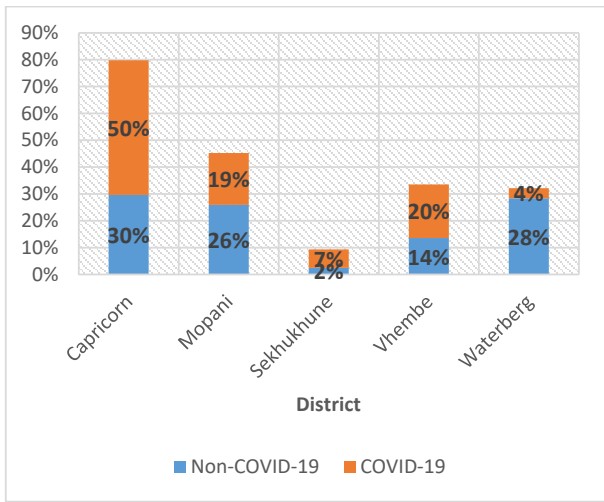

**Figure 2.** Prevalence of mortality among individuals with laboratory-confirmed SARS-CoV-2 according to districts (*p*–value < 0.001).

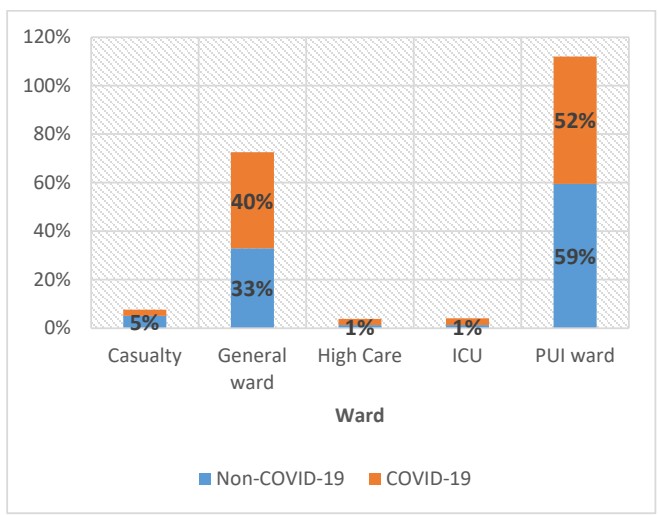

**Figure 3.** Prevalence of mortality among individuals with laboratory-confirmed SARS-CoV-2 according to the ward (*p*–value = 0.815).

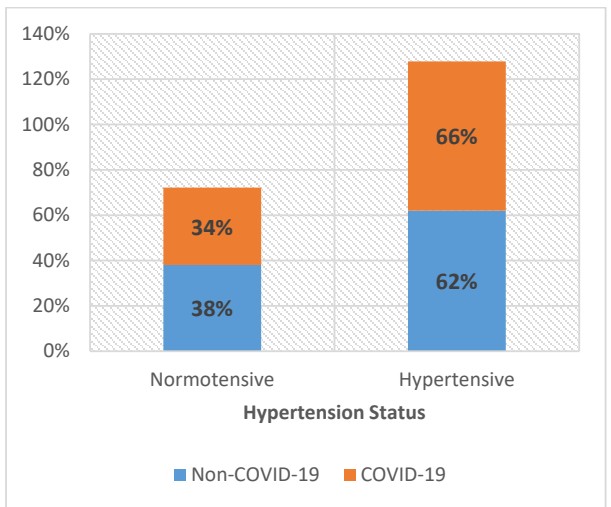

**Figure 4.** Prevalence of mortality among individuals with laboratory-confirmed SARS-CoV-2 according to hypertension status (*p*-value = 0.510).

The results of the hierarchical logistic regression for hypertension status undertaken on hospitalised patient mortality with laboratory-confirmed SARS-CoV-2 showed a significant association between HIV/AIDS positive status and diabetes mellitus with hypertension using model 1 (shown in Table 2). In model 2, sociodemographics were added to model 1 and there was a non-significant association between sociodemography and hypertension; however, a significant association was only observed between diabetes mellitus with hypertension (OR = 5.04 (3.22; 7.89); *p*-value < 0.001). In model 3, case-severity and patient-related factors were added to model 2, and a significant association between HIV/AIDS status (OR = 0.42 (0.18; 0.99); *p*-value = 0.046), diabetes (OR = 3.19 (1.81; 5.62); *p*-value < 0.001) and chronic treatment (OR = 8.50 (4.71; 15.34); *p*-value < 0.001) with hypertension was found. On the other hand, those patients aged 60 or above were significantly associated with hypertension (with a prevalence of 73.4%) at a 10% significant level among hospitalised patient mortality with laboratory-confirmed SARS-CoV-2.

**Table 2.** Prevalence of hypertension of all the patients admitted with COVID-19 in Limpopo Province.

| Variables | Normotensive | Hypertensive 586 (64%) | Model 1: OR (95% CI) | Model 2: OR (95% CI) | Model 3: OR (95% CI) | *p*-Value |
|---|---|---|---|---|---|---|
| **Comorbid conditions** | | | | | | |
| HIV/AIDS | 40 (46.0%) | 47 (54.0%) | **0.46 (0.23; 0.91)** | 0.64 (0.30; 1.34) | **0.42 (0.18; 0.99)** | 0.046 |
| TB | 17 (70.8%) | 7 (29.2%) | 0.35 (0.09; 1.33) | 0.44 (0.11; 1.84) | 0.34 (0.07; 1.59) | 0.171 |
| COPD | 5 (45.5%) | 6 (54.5%) | 1.64 (0.34; 7.8) | 1.09 (0.22; 5.42) | 1.45 (0.26; 7.99) | 0.672 |
| Diabetes mellitus | 52 (16.1%) | 271 (83.9%) | **4.73 (3.09; 7.2)** | **5.04 (3.22; 7.89)** | **3.19 (1.81; 5.62)** | <0.001 |
| Asthma | 8 (32.0%) | 17 (68.0%) | 0.99 (0.34; 2.93) | 1.13 (0.37; 3.48) | 0.59 (0.16; 2.19) | 0.437 |
| Obesity | 11 (19.3%) | 46 (80.7%) | 1.74 (0.71; 4.30) | 2.46 (0.90; 6.71) | 1.58 (0.48; 5.27) | 0.456 |
| Cancer | 5 (29.4%) | 12 (70.6%) | 3.14 (0.93; 10.52) | 3.23 (0.86; 12.07) | 4.15 (0.45; 38.24) | 0.209 |
| **Gender** | | | | | | |
| Female | 128 (32.7%) | 263 (67.3%) | | Reference | Reference | |
| Male | 114 (35.8%) | 204 (64.2%) | | 0.97 (0.63; 1.49) | 0.94 (0.54; 1.63) | 0.848 |
| **Age** | | | | | | |
| 0–29 | 3 (75.0%) | 1 (25.0%) | | Reference | Reference | |
| 30–39 | 25 (78.1%) | 7 (21.9%) | | 0.50 (0.03; 8.47) | 0.84 (0.04; 18.66) | 0.913 |
| 40–49 | 31 (51.7%) | 29 (48.3%) | | 1.01 (0.07; 13.97) | 2.244 (0.13; 36.74) | 0.571 |
| 50–59 | 48 (45.3%) | 58 (54.7%) | | 2.07 (0.16; 27.42) | 3.791 (0.24; 58.25) | 0.339 |
| ≥ 60 | 135 (26.6%) | 372 (73.4%) | | 4.96 (0.39; 62.95) | **10.66 (0.73; 154.3)** | 0.082 |
| **Case severity** | | | | | | |
| Mild | 14 (33.3%) | 28 (66.7%) | | | Reference | |
| Moderate | 26 (27.1%) | 70 (72.9%) | | | 2.36 (0.46; 12.07) | 0.302 |
| Severe | 194 (35.9%) | 347 (64.1%) | | | 2.09 (0.45; 9.68) | 0.346 |
| **Patient-related factors** | | | | | | |
| On chronic treatment | 82 (22.6%) | 281 (77.4%) | | | **8.50 (4.71; 15.34)** | <0.001 |
| Respiratory distress | 59 (38.3%) | 95 (61.7%) | | | 0.88 (0.29; 2.76) | 0.837 |

Bold significant at 5% level; Model 1: comorbid conditions; Model 2: comorbid conditions and sociodemographic; Model 3: comorbid conditions and sociodemographic, case-severity and patient-related factors.

## 4. Discussion

According to the current literature on COVID-19 documents, several factors are associated with the prognosis of infected people. It was found that comorbidities and older age are some of the risk factors for a worse prognosis of COVID-19 [8,9,14]. The most common underlying condition in people with severe COVID-19 was shown to be hypertension [8,15,16]. Therefore, this study determined the prevalence of hypertension in hospitalised COVID-19 patients who died and investigated the factors associated with

hypertension among the study population located in Limpopo Province, South Africa. The study showed a high rate of mortality in COVID-19 hospitalised patients of older age, amongst patients in PUI wards and patients presenting with cough. It could be because the patient did not show up on time (while in respiratory distress), which caused them to die in the emergency room or while waiting for test results in the PUI ward. Furthermore, the study revealed that the most common underlying diseases in hospitalised COVID-19 patients who died were respiratory distress, diabetes mellitus and hypertension. Our findings also confirm prior research that found diabetes and hypertension to be common among COVID-19 patients [8,9,15,16]. The National Department of Health defines a suspected COVID-19 case as any person presenting with an acute (≤14 days) respiratory tract infection or other clinical illness compatible with COVID-19 or an asymptomatic person who is in close contact with a confirmed case [17]. For symptomatic suspected COVID-19 cases, a broad differential diagnosis should be considered. Ninety-two percent (92%) of all suspected COVID-19 cases referred to hospitals in this study were laboratory-confirmed SARS-CoV-2 cases.

We found no significant association between COVID-19 case severity and hypertension in hospitalised patients who died. Our findings also lend support to previous studies that indicated that hypertension does not affect the outcome of COVID-19 [17]. These findings are in contradiction to other studies that found a significant association between the death of COVID-19 patients and hypertension [12,16–20]. It is still unknown how COVID-19 infection causes underlying cardiovascular conditions to worsen. However, studies suggest a connection between COVID-19 and the renin-angiotensin system (RAS) [20,21]. The activation of RAS, among other important pathophysiological hypertension mechanisms, possibly has an impact on COVID-19. The SARS-CoV-2 also targets angiotensin-converting enzyme 2 (ACE2) receptors as entry points to human host cells [21,22]. No hypertension treatment protocols were provided in the current study, which could have had a significant impact on the outcomes of COVID-19 patients. The comparison of the current findings with similar studies in other populations is challenging due to the lack of therapeutic data. Further investigation is needed in this area to study the prognostic factors among COVID-19 patients with underlying hypertension to anticipate probable outcomes.

In the current study, hypertension was shown to be significantly associated with HIV/AIDS among hospitalised COVID-19 patients who died. Several studies have demonstrated that inflammatory markers of HIV-related chronic immune activation are associated with hypertension [23]. According to the findings of Jassat et al., among laboratory-confirmed SARS-CoV-2 patients admitted to a hospital, persons living with HIV (PLWH) have a lower chance of getting tested for hypertension [15,24]. Furthermore, AdultPLWH with CD4 ≤ 200 cells/mm$^3$ had 44% lower odds of hypertension compared to adults with CD4 > 200 cells/mm$^3$ [24].

In the present study, hypertension was significantly associated with diabetes among hospitalised COVID-19 patients who died. Diabetes patients were three times more likely to have hypertension compared to non-diabetes people. This is in agreement with the previous studies, which reported that the odds of hypertension were higher with diabetic participants, particularly among women in rural communities in South Africa [25,26]. Furthermore, in the study, it was observed that people of older age (>60 years of age) were ten times more likely to be hypertensive compared to younger people (≤60 years of age). The proportion of people with hypertension increased with advancing age [25–27].

A significant 68% of hospitalised patients who had asthma also reported having hypertension. A study conducted among Korean adults found that hypertension and diabetes are potential associated factors for the development of asthma [28,29]. Having both asthma and hypertension worsens overall health and raises the chance of mortality if they are not well controlled [29]. However, in this study, no significant association was observed between asthma and hypertension. Furthermore, no significant link was found between TB, cancer and hypertension in the study. Findings from systematic reviews also lend support to the findings of our study, in that they found no evidence to support an

association between TB, cancer and hypertension [30,31]. Obesity is a major risk factor for hypertension, and the combination of obesity and hypertension is a leading cause of cardiovascular risk [32,33]. However, in this study on the mortality of hospitalised COVID-19 patients, there was no significant association between obesity and hypertension.

The limitations of this study are worth noting. The first limitation concerns the use of secondary data from the Limpopo Department of Health. The available data were not collected to address the study research question. Some variables were not available for analysis such as dietary, physical activity status and consumption of tobacco and alcohol which might have resulted in residual confounding when the variables are critical to account for in the study analysis. Secondly, the study's researchers were not involved in data collecting. As a result, they are unlikely to be aware of study-specific nuances or flaws in the data collection method that may be crucial in interpreting certain variables in the dataset. Lastly, the possibility of incomplete or inaccurate healthcare data must be acknowledged because they were derived from patient files. Despite these noted limitations, our study adds knowledge to the body of literature on the prevalence of hypertension among hospitalised COVID-19 patients who died in South African hospitals and its association with the COVID-19 case severity.

**5. Conclusions**

A high prevalence of hypertension was observed in hospitalised COVID-19 patients. Among the hospitalised COVID-19 patients who died, prominent risk factors for hypertension were old age, diabetes, HIV/AIDS and being on chronic therapy. There was no evidence to establish a link between hypertension and COVID-19 case severity. No hypertension treatment protocols were provided, which could have had a significant impact on the outcomes of COVID-19 patients. The comparison of the current findings with similar studies in other populations is challenging, due to the lack of therapeutic data. More cohort and systematic studies are needed to determine whether there is a link between hypertension and COVID-19 case severity. Awareness and understanding of such an association would be of relevance to health care providers, in order to prevent and reduce the risk of COVID-19 mortality.

**Author Contributions:** Conceptualisation, P.M.M.; methodology, P.M.M. and M.E.S.-S.; software, P.M.M.; validation, P.M.M., formal analysis, P.M.M.; resources, P.M.M. and M.E.S.-S.; data curation, M.E.S.-S. and N.J.R.; writing—original draft preparation, P.M.M.; writing—review and editing, P.M.M., S.F.M., T.G.T. and M.E.S.-S.; visualisation, P.M.M.; project administration, M.E.S.-S. All authors have read and agreed to the published version of the manuscript.

**Funding:** This research received no external funding.

**Institutional Review Board Statement:** The study was conducted in accordance with the Declaration of Helsinki and approved by the Institutional Review Board (or Ethics Committee) of the University of Limpopo (protocol code TREC/293/2021: IR-11/2021).

**Informed Consent Statement:** Patient consent was waived because the study used secondary data from the Limpopo Department of Health.

**Data Availability Statement:** Not applicable.

**Acknowledgments:** The authors would like to thank the Limpopo Department of Health for granting access to COVID-19 mortality data. The clinical managers and nursing service managers who coordinated each hospital audit team.

**Conflicts of Interest:** The authors declare no conflict of interest.

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
