# Peer review of "Risk Factors for Hypertension in Hospitalised Patient Mortality with Laboratory-Confirmed SARS-CoV-2: A Population-Based Study in Limpopo Province, South Africa"

_2673-527X, doi:10.3390/jor2030013_

Round 1

Reviewer 1 Report

The manuscript is well written and the data organization is good, however, this study is inclusive because of low population size and finding against the previous reports where hypertension has been linked with covid-19 death.

Author Response

The authors concur with the reviewer’s comment.  Despite these noted limitations, the authors believe that the study adds knowledge to the body of literature on the prevalence of hypertension among hospitalised COVID-19 patients who died in South African hospitals and its association with the COVID-19 case severity. Since there no evidence to establish a link between hypertension and COVID-19 case severity. The study make a call for more cohort and systematic studies to determine whether there is a link between hypertension and COVID-19 case severity.

Thank you for all the comments and recommendations. The opinions were really helpful to eliminate misunderstanding and several major flows, and to make the manuscript more accurate and valuable

Reviewer 2 Report

The submitted retrospective analysis according to a population-based study in Limpopo province , South Africa was clearly presented and thoroughly analysed. The figures and tables are displayed beautifully. However, there are some points that have to be stressed out: 

a) Please do define the Title of Table 2. 

b) Under the Discussion-Part --> First Sentence , would be better in this other version like: "According to the current literature of Covid-19 ...."

c)Under the Limitations-Part --> Please do define the critical third variables , which were not assessed in the analysis and could have been potential biases. This would make it more transparent to cope with.  

Author Response

a) Please do define the Title of Table 2. 

The authors added a caption for Table 2 as “Table 2. Prevalence of hypertension of all the patients admitted with COVID-19 in Limpopo Province

b) Under the Discussion-Part --> First Sentence , would be better in this other version like: "According to the current literature of Covid-19 ...."

The authors amended the first sentence as suggested.

c)Under the Limitations-Part --> Please do define the critical third variables , which were not assessed in the analysis and could have been potential biases. This would make it more transparent to cope with.  

The authors rephrased the statement as “Some variables were not available for analysis such as dietary; physical activity status; consumption of tobacco and alcohol; which might have resulted in residual confounding when the variables are critical to account for in the study analysis”

Thank you for all the comments and recommendations. The opinions were really helpful to eliminate misunderstandings and several major flows and to make the manuscript more accurate and valuable

Reviewer 3 Report

Current study demonstrated that no evidence to establish a link between hypertension and COVID-19 case severity. I like to give the following comments.

1.      The disrupted hypertension treatment has been indicated as helpful in COVID-19 pandemic. The association between hypertension and COVID-19 needs to introduce in detail.

2.      COVID-19 patients between the first (16 March 16 2020 to 31 October 2020) and second (1 November 2020 to 31 March 2021) pandemic waves were compared in current study. Is the type of Omicron same during October 2020 and March 2021?

3.      Patients with hypertension in Table 1 were 586 and it was 235 in Table 2 without title. Why?

4.      Please indicate Table2 in clear.

5.      Cases of laboratory-confirmed SARS-CoV-2 must distinguish in clear.

6.      Patients aged 60 or above were significantly associated with diabetes as shown in Table 2. What is the prevalence of mortality in these cases?

7.      Results belonged to negative. Therefore, factors may involve in current findings shall be discussed in detail.

8.      Case number is one of the limitations.

9.      Conclusion needs the potential reason(s) to explain the current findings.

Author Response

  1. The disrupted hypertension treatment has been indicated as helpful in COVID-19 pandemic. The association between hypertension and COVID-19 needs to introduce in detail.

The authors tried to emphasize why other study got association between hypertension and COVID-19  as “It is still unknown how COVID-19 infection causes underlying cardiovascular conditions to worsen. However, studies suggest a connection between COVID-19 and the renin-angiotensin system (RAS) [20,21]. The activation of RAS, among other important pathophysiological hy-pertension mechanisms, possibly has an impact on COVID-19. The SARS-CoV-2 also targets angiotensin-converting enzyme 2 (ACE2) receptors as entry points to human host cells [21,22]. Further investigation is needed in this area to study the prognostic factors among COVID-19 patients with underlying hypertension to anticipate probable outcomes”.

  1. COVID-19 patients between the first (16 March 16 2020 to 31 October 2020) and second (1 November 2020 to 31 March 2021) pandemic waves were compared in current study. Is the type of Omicron same during October 2020 and March 2021?

The authors concur that while the type of Omicron did not remain constant throughout the waves, the percentage of hypertension was nearly constant (wave 1=65% and wave 2=68%).

  1. Patients with hypertension in Table 1 were 586 and it was 235 in Table 2 without title. Why?

The authors acknowledge that as a typing error and it was rectified as 586 (64%) and again caption was added to Table 2 as “Table 2. Prevalence of hypertension of of all the patients admitted with COVID-19 in Limpopo Province

  1. Please indicate Table2 in clear.

The authors added caption for Table 2 as “Table 2. Prevalence of hypertension of of all the patients admitted with COVID-19 in Limpopo Province

  1. Cases of laboratory-confirmed SARS-CoV-2 must distinguish in clear.

The authors added a statement as “In Figures, non-COVID-19 patients were viewed as highly suspicious of COVID-19 by medical professionals, but non-lab-confirmed SARS-CoV-2 patients. Either awaiting test results or had not yet been tested. Of all the patients admitted to referral hospitals in Lim-popo Province with clinical syndromes highly suspicious of COVID-19 (969), only 887 (92%) were laboratory-confirmed SARS-CoV-2

  1. Patients aged 60 or above were significantly associated with diabetes as shown in Table 2. What is the prevalence of mortality in these cases?

Patients aged 60 or above were significantly associated with hypertension and with a prevalence of 73.4%

  1. Results belonged to negative. Therefore, factors may involve in current findings shall be discussed in detail.

The authors tried to add a statement to support their negative results as “No hypertension treatment protocols were provided, which could have had a significant impact on the outcomes of COVID-19 patients. The comparison of the current findings with similar studies in other populations is challenging due to the lack of therapeutic data

  1. Case number is one of the limitations.

The authors acknowledged incomplete or inaccurate healthcare data because it was derived from patient files

  1. Conclusion needs the potential reason(s) to explain the current findings.

The authors added a statement to support the results “No hypertension treatment protocols were provided, which could have had a significant impact on the outcomes of COVID-19 patients. The comparison of the current findings with similar studies in other populations is challenging due to the lack of therapeutic data

Thank you for all the comments and recommendations. The opinions were really helpful to eliminate misunderstanding and several major flows, and to make the manuscript more accurate and valuable

Round 2

Reviewer 3 Report

It has been improved in a good way.